# Evaluating Implicit Generative Models With Large Samples

**Ishaan Gulrajani, Colin Raffel, Luke Metz**
Google Brain
{igul,craffel,lmetz}@google.com

## Abstract

We study the problem of evaluating a generative model using only a finite sample from the model. For many common evaluation functions, generalization is meaningless because trivially memorizing the training set attains a better score than the models we consider state-of-the-art. We clarify a necessary condition for an evaluation function not to behave this way: estimating the function must require a large sample from the model. In search of such a function, we turn to parametric adversarial divergences, which are defined in terms of a neural network trained to distinguish between distributions: as we make the network larger, the function is less easily minimized by memorizing the training set. We implement a reliable evaluation function based on these ideas, validate it experimentally, and show models which achieve better scores than memorizing the training set.

## 1 Generalization in Implicit Models

We begin this section by discussing what it means for a model to generalize with respect to a given evaluation function. We then clarify the tradeoff between an evaluation function's ability to permit meaningful generalization and its ability to be estimated from a finite sample. Finally, we propose to mitigate this tradeoff through evaluation functions which consider a large sample from the model.

### 1.1 Models Should Generalize Beyond the Training Set

We take $D(\hat{p}_{\text{test}}, q)$ to be our evaluation function, which we minimize $D(\hat{p}_{\text{test}}, q)$ by some indirect process like optimizing a surrogate training loss. Intuitively, generalization means that the training process succeeds in making $D(\hat{p}_{\text{test}}, q)$ small.

Let $\hat{p}_n$ be an empirical distribution of $n$ points from $p$. For some $D$, we have that $D(\hat{p}_{\text{test}}, \hat{p}_n) \rightarrow D(\hat{p}_{\text{test}}, p)$ as $n \rightarrow \infty$. This means that even trivially memorizing the training set generalizes to some extent under $D$, since it minimizes $D(\hat{p}_{\text{test}}, q)$ to some extent. However, such memorization is presumably not a useful solution in terms of our final task, so we propose that in order for generalization of a model to be considered meaningful, it should at least generalize beyond the level of the training set. Specifically:

**Definition 1.** *A model $q$ generalizes meaningfully if $D(\hat{p}_{\text{test}}, q) < D(\hat{p}_{\text{test}}, \hat{p}_{\text{train}})$.*

A similar criterion is discussed in Cornish et al. (2018).

### 1.2 The Tradeoff Between Fast Convergence and Meaningful Generalization

The intuition in this section is that an evaluation function which can be estimated with only a small sample from $q$ can be fooled by $q$ which memorizes a small sample, but on the other hand a small sample from is all we have from $q$.

Whether a model generalizes depends on the choice of evaluation function $D$. For instance, if $D$ is such that $D(\hat{p}_{\text{test}}, \hat{p}_{\text{train}})$ is already very small, then finding a model which outperforms memorization of the training set might be very difficult. Arora et al. (2017) prove that the parametric adversarial divergences commonly used to train GANs have this property for modestly large $\hat{p}_{\text{train}}$. More gener-

ally, any $D$ for which there exists a "good" estimator using only a finite sample from $q$ must behave like this. Specifically:

**Example 1.** *If an evaluation function $D(\hat{p}_{\text{test}}, q)$ can be estimated with error at most $\epsilon$ using only a size-$m$ sample from $q$, and $\hat{p}$ is an empirical version of $p$ with $m^2$ points, then $|D(\hat{p}_{\text{test}}, \hat{p}) - D(\hat{p}_{\text{test}}, p)| \leq 2\epsilon$.*

This follows easily from Gretton et al. (2012), who point out that if $q$ is a size-$m^2$ sample from $p$, and $\hat{q}$ is a size-$m$ sample from $q$, then $\hat{q}$ is quite likely to also be a valid sample from $p$, making $p$ and $q$ indistinguishable by any test which relies only on $\hat{q}$.

However, in practice we only have access to a finite sample from $q$.As we consider $D$ with faster-converging estimators, we ignore the difference between a distribution and its empirical counterpart; as we consider $D$ with slower-converging estimators, the error makes it less likely that our estimate correlates with a quantity we care about. Computational complexity also prohibits us from considering large samples for some $D$.

## 1.3 Mitigating the Tradeoff

We might hope to mitigate the tradeoff problem by taking advantage of the fact that we usually have access to several orders of magnitude more sample points from $q$ than $\hat{p}_{\text{train}}$. For instance, the CIFAR-10 training set contains 50,000 images, but we can draw a sample of 50 million images from a typical GAN generator within a reasonable computational budget. Therefore if we consider an evaluation function for which estimation requires as many sample points from $q$ as possible, but fewer than 50 million, we can hope that memorizing $\hat{p}_{\text{train}}$ might be a worse solution under such a divergence than our best models.

## 2 A Large-Sample Evaluation Function

In this section we propose to use a parametric adversarial divergence (Huang et al., 2017) as an evaluation function for generative modeling.

Parametric adversarial divergences have unique advantages at sample sizes on the order of millions of observations. First, unlike with the Inception Score or FID, the sample complexity can be tuned by changing the capacity of the underlying neural network. Smaller networks require fewer samples to train, but detect memorization less effectively (Arora et al., 2017). The second advantage is computational: the FID and kernel-based methods like MMD both have runtime quadratic (or worse) in the number of sample points, which makes evaluation expensive when considering millions of samples. In contrast, parametric divergences have runtime linear in the number of generator sample points (since we can draw arbitrarily many generator sample points, each forward or backward pass considers a new batch of sample points).

### 2.1 Our Implementation

Evaluating a parametric adversarial divergence involves training a neural network, which means that implementation is nontrivial compared to most evaluation functions and minor changes to the implementation can result in an entirely different divergence function. Therefore to support experimentation and direct comparison of results, we develop a standardized implementation of a specific parametric adversarial divergence which we call the *WGAN-GP divergence*, which we will release alongside this paper. In our implementation we tune architecture and training hyperparameters to achieve the following: Reasonable computational cost (our code runs in about an hour on a P100 accelerator), low variance between runs due to stochasticity from the neural network training process, a divergence measure which doesn't saturate easily when the distributions are easily discriminable (Arjovsky & Bottou, 2017), and enough capacity in the network and training process to penalize memorization at the level of a typical training set.

## 3 EXPERIMENTS

### 3.1 VARIANCE BETWEEN RUNS

A necessary property of any useful evaluation function is that running it multiple times yields the same (or similar) result. We train a GAN on CIFAR-10 and run our evaluation function $D(\hat{p}_{\text{train}}, q)$ $n = 50$ times. The resulting values have a mean of 5.51 and standard deviation of 0.03, which is approximately 0.5% of the mean. We conclude that our divergence is reliable across runs.

### 3.2 TRACKING CONVERGENCE OF TRAINING

We train a WGAN-GP for 100,000 iterations, save model checkpoints every 1000 iterations. We evaluate each checkpoint using our divergence and plot the results in Figure 1. We find that our divergence reflects the convergence of GAN training in this case.

### 3.3 COMPARING EVALUATION FUNCTIONS' ABILITY TO PENALIZE MEMORIZATION

For several evaluation functions, we test how many sample points from the true distribution are needed to attain a better score than a well-trained GAN model.

We begin by training a WGAN-GP model $q$ on the $32 \times 32$ ImageNet dataset (Oord et al., 2016). We estimate each evaluation function $D(\hat{p}_{\text{test}}, q)$ (denoting the estimate as $\hat{D}$) using samples from $\hat{p}_{\text{test}}$ and $q$, where the sample size is the same as commonly used for that evaluation function. We similarly estimate $\mathbb{E}_{\hat{p}_n}[D(\hat{p}_{\text{test}}, \hat{p}_n)]$, where $\hat{p}_n$ is a random $n$-point subset of $\hat{p}_{\text{train}}$, for many values of $n$. In Table 1 we report the smallest $n$ for which $\hat{D}(\hat{p}_{\text{test}}, q) < \mathbb{E}_{\hat{p}_n}[\hat{D}(\hat{p}_{\text{test}}, \hat{p}_n)]$ for each evaluation function $D$. We find that WGAN-GP divergence assigns relatively more importance to the difference between $p$ and $\hat{p}_n$, so generalization in the sense of Definition 1 is more likely to be achievable. These numbers are slightly optimistic because we train the GAN on the *test* split of the dataset, so that to the extent it overfits, it is not penalized for doing so in this comparison.

### 3.4 THE EFFECT OF A SMALL TEST SET

We train 64 GAN models on ImageNet with randomly-chosen hyperparameters on the training set and evaluate their WGAN-GP divergence with respect to a small and a large test set. We plot the results in Table A. We observe that the two values are strongly correlated over this set of models (up to the level of noise introduced by the optimization process), suggesting that it might be safe to use a small test set in this setting.

### 3.5 EVALUATING MODELS AGAINST WGAN-GP DIVERGENCE

Having analyzed our evaluation technique, we now use it to evaluate several generative models. We train 3 models: a PixelCNN++ (Salimans et al., 2017), a ResNet VAE with Inverse Autoregressive Flow (IAF) Kingma et al. (2016), and a DCGAN (Radford et al., 2015) trained with the WGAN-GP objective (Gulrajani et al., 2017). We list results in Table 2. We observe that the GAN achieves a better score than either memorizing the training set or either maximum-likelihood model (which makes sense, because its training objective is much closer to the evaluation function than maximum-likelihood).

#### 3.5.1 OVERFITTING IN GANS

We experimentally show evidence of overfitting in GANs using WGAN-GP divergence. We train a large, but not unreasonably so (18M parameters), GAN (using the WGAN-GP algorithm) on a 50,000-image subset of $32 \times 32$ ImageNet and measure three WGAN-GP divergences every 2000 iterations over the course of training. The first divergence is with respect to a held-out test set of $10,000$ images. The second is with respect to another independent test set of equal size, to verify that the divergence has negligible variance with respect to the random choice of test set images. The third is with respect to a $10,000$-image subset of the training set, where we use a subset of equal size to the test sets to eliminate bias from the size of the dataset. We plot the results in Figure 3.

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

## A  FIGURES AND TABLES

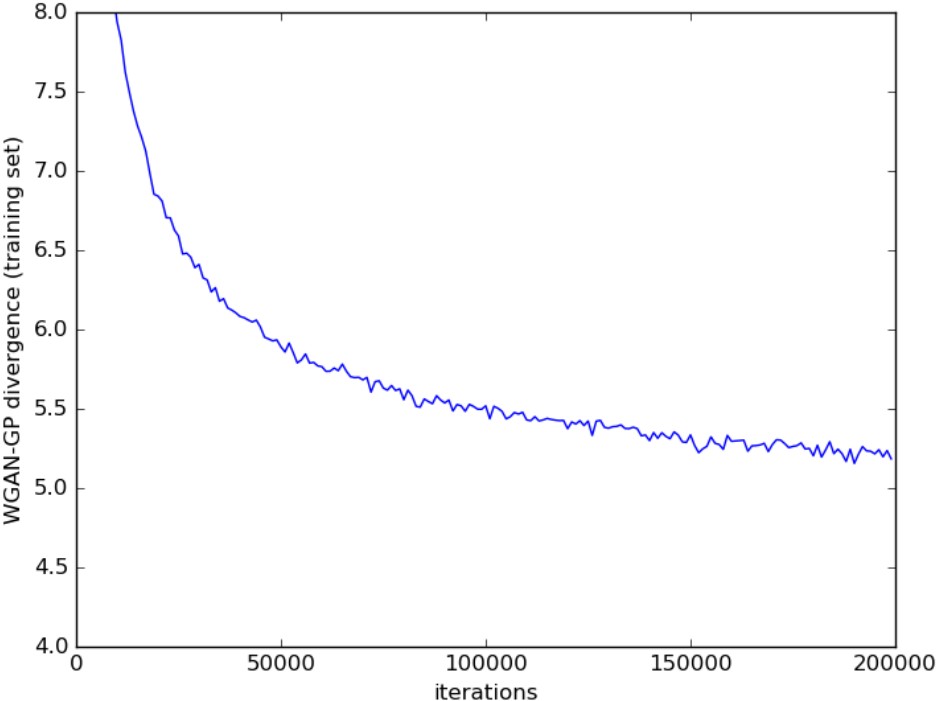

Figure 1: WGAN-GP divergence evaluated every 1000 iterations over the course of GAN training.

Table 1: For a fixed model $q$ and different objectives $D$, the sample size used to estimate $\hat{D}$ and the lower bound on $n$ at which $\hat{D}(\hat{p}_{\text{test}}, q) < \mathbb{E}_{\hat{p}_n}[\hat{D}(\hat{p}_{\text{test}}, \hat{p}_n)]$. A larger bound on $n$ means generalization beyond the training set is easier.

| EVALUATION FUNCTION | SAMPLE SIZE | $\hat{D}(\hat{p}_{\text{test}}, q) < \mathbb{E}_{\hat{p}_n}[\hat{D}(\hat{p}_{\text{test}}, \hat{p}_n)]$ |
|---|---|---|
| INCEPTION SCORE | 5000 | $n > 32$ |
| FID | 50,000 | $n > 1024$ |
| WGAN-GP DIVERGENCE (SMALL) | 25M | $n > 32{,}768$ |
| WGAN-GP DIVERGENCE | 25M | $n > 1M$ |

Table 2: Evaluation of different models on CIFAR-10 by test set WGAN-GP divergence. In particular, the WGAN-GP attains a lower value of the test divergence than memorization of the training set.

| METHOD | $D_{\text{WGAN-GP}}(\hat{p}_{\text{test}}, q)$ |
|---|---|
| PIXELCNN++ | 16.17 |
| IAF VAE | 18.11 |
| WGAN-GP | 12.97 |
| TRAINING SET | 14.62 |

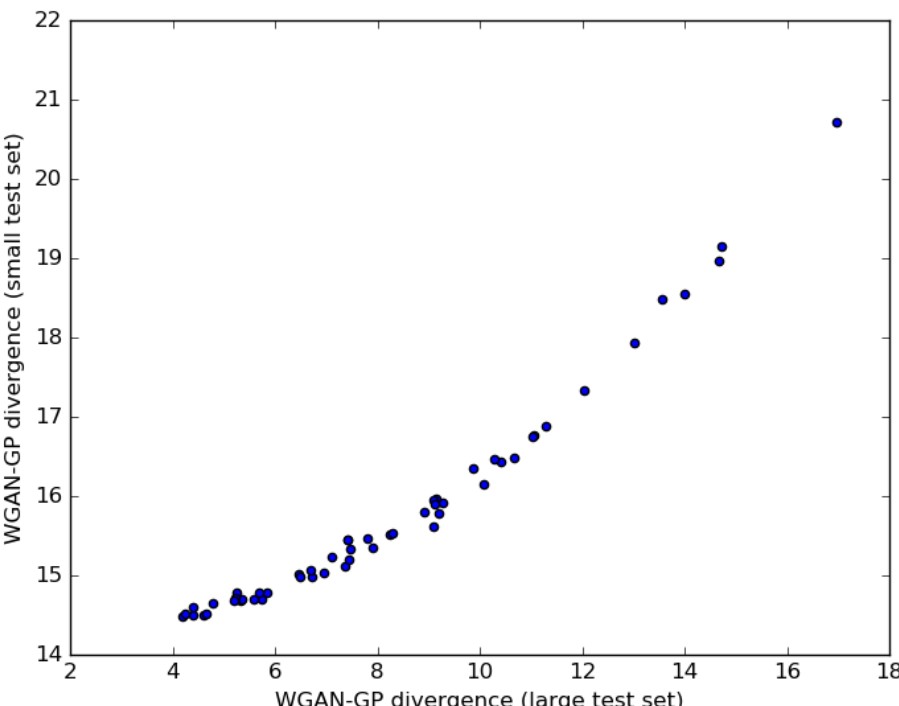

Figure 2: WGAN-GP divergence evaluated with respect to a small test set is a biased estimate of the true value, but correlates well for typical GAN models.

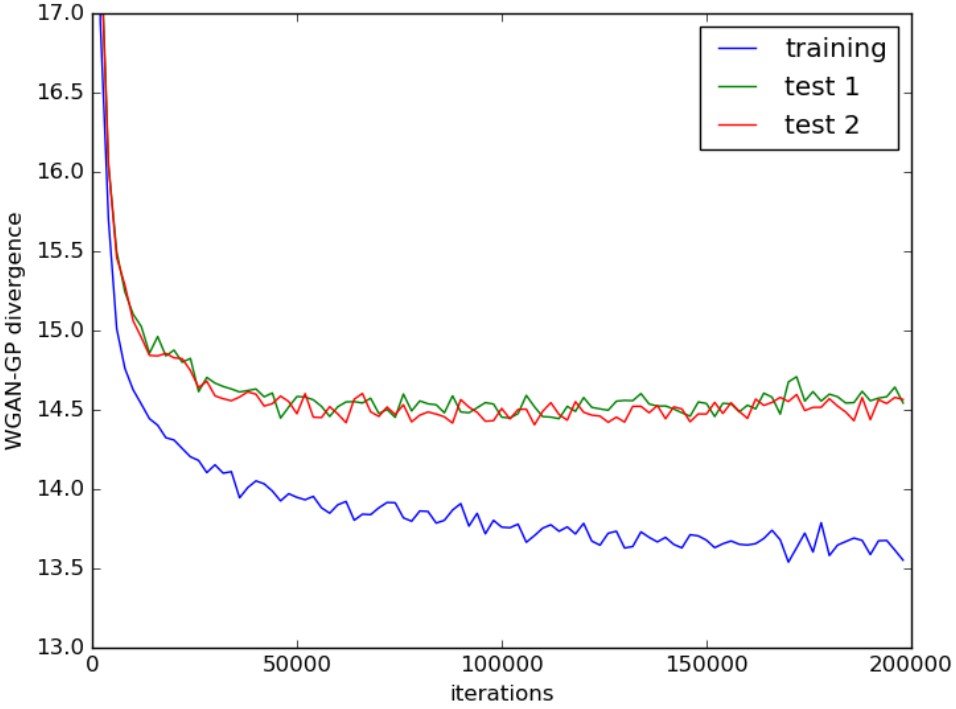

Figure 3: WGAN-GP divergence reveals substantial overfitting in a GAN with a typical architecture and dataset.

