# OpenReview forum: "Evaluating Implicit Generative Models With Large Samples"
_ICLR.cc/2018/Workshop — Reject_

### Official Review · AnonReviewer2 · 2018-03-05
**Interesting initial contribution**

**Rating:** 7
**Confidence:** 5

**Review:**

This paper provides an interesting initial version of a proposal for a very important topic, and is clearly worthy of inclusion in the workshop.

There remain many practical and theoretical considerations to be made for this method: for example, does it "unfairly" favor models based on similar architectures? Though the variance is low, the bias is clearly high (from e.g. Figure 2) -- how big of a deal is this? In particular, does this bias make it difficult to compare estimates between different distributions (which is what you really care about)? How sensitive are the evaluations to minor changes in the architecture / training procedure / etc? Are the gains in evaluation worth the large amount of additional computational required over an Inception score / FID-type evaluation? These and others are all important questions to grapple with for this type of evaluation method, but beyond the scope of a three-page workshop abstract.

One minor correction: you say in section 2 that the FID and MMD-like methods "have runtime quadratic (or worse) in the number of sample points." This is not at all true for the FID, whose default estimator has runtime O(n d^2 + d^3) for n the number of sample points and d the dimension. Additionally, although the "default" estimator for the MMD does take time quadratic in the samples, there is also a linear-time estimator given by Gretton et al. (2012), a generalization that can still be chosen to be linear-time by Zaremba et al. (NIPS 2013), and a different linear-time estimator of a closely related distance by Chwialkowski et al. (NIPS 2015) / Jitkrittum et al. (NIPS 2016).

---

### Official Review · AnonReviewer3 · 2018-03-10
**Interesting idea but lacks of new / claim-supporting quantitative results**

**Rating:** 5
**Confidence:** 3

**Review:**

This paper tries to quantitatively argue that the parametric adversarial divergences are good metrics to evaluate implicit generative models with large samples. But i have concerns on the new results presented in this work. Tracking the convergence of training (Figure 1) and overfitting issues (Figure 3) have already been explored by the original WGAN work (Figure 3) and the follow-up WGAN-GP work (Figure 5).

For the analysis in Table 1, I do not think it would directly relate to Definition 1. The divergence D in Definition 1 is the same for both sides of the inequality. But the trained parametric adversarial divergences on different sides of the inequality in Table 1 are trained on different datasets suggesting they are in high probability different divergences. Even though different training of the network on the same pair of (test data, generated data) results consistent evaluations, i do not believe this could be extended to different pairs of data.

One of the emphasis of the work is to use "large sample" from the generative model for evaluation. The result shown in Figure 2 suggests that "it might be safe to use a small test set" from the data distribution. My understanding is that this comparison is based on fixed number of samples from the generative model. What we usually deal with in practice is a fix number of test samples from the data distribution. Wouldn't this result also suggest that it might also be safe to a use a small set from the generative model, which would undermine the usage of "large sample".

---

### Official Review · AnonReviewer1 · 2018-03-10
**Maybe it contains some valuable empirical insights, but the paper currently doesn't work**

**Rating:** 2
**Confidence:** 3

**Review:**

There's something fundamental that I don't understand about this paper. The premise is that, for many evaluation metrics, memorising the training set yield good generalisation performance. I just can't make sense of that statement. What are those evaluation metrics? What's the exact technical statement you are making? The closest to an explanation that I can find in the paper is the following paragraph:

  Let pˆn be an empirical distribution of n points from p. For some D, we have that D(ˆptest, ˆp) →
  D(ˆptest, p) as n → ∞. This means that even trivially memorizing the training set generalizes to
  some extent under D, since it minimizes D(ˆptest, q) to some extent

That's extremely hand-wavy and clearly insufficient motivation for the paper. A concept such as generalisation (or just convergence) is pretty subtle and I'd suggest the authors review statistical learning theory and take a more principled approach to this issue. How do you even define D(ˆptest, pˆn) if the two empirical distributions have different support? That's what you need learning for, to go from ˆp_n to a function on the whole input space including ˆp_test. But there's no mention of a learning algorithm anywhere.

I think that the authors are implicitly thinking about the GAN case and probably have an interesting good point to make from an empirical perspective, but they need to focus on the specific issue they have in mind and be much more precise instead of talking about key concepts such as generalisation in such a broad-stroked manner.

Minor points
* Lacking definitions: Just based on the first few lines: what is an implicit model in this context? what are p_test and q? I can have educated guesses, but I shouldn't have to.
* Writing is "amateur" in general and doesn't read like a proper paper

---

### Decision · Program_Chairs · 2018-03-20
**ICLR 2018 Workshop Acceptance Decision**

**Decision:**

Reject

**Comment:**

Based on the reviews, this paper has not been accepted for presentation at the ICLR workshop. However, the conversation and updates can continue to appear here on OpenReview.